# Negative Impact and Positive Value of Caregiving in Spouse Carers of Persons with Dementia in Sweden

**DOI:** 10.3390/ijerph19031788

**Published:** 2022-02-04

**Authors:** Marcus F. Johansson, Kevin J. McKee, Lena Dahlberg, Martina Summer Meranius, Christine L. Williams, Lena Marmstål Hammar

**Affiliations:** 1School of Health and Welfare, Dalarna University, 791 88 Falun, Sweden; kmc@du.se (K.J.M.); ldh@du.se (L.D.); lma@du.se (L.M.H.); 2Aging Research Center, Karolinska Institutet & Stockholm University, Tomtebodavägen 18A, 171 65 Solna, Sweden; 3School of Health, Care and Social Welfare, Mälardalen University, 721 23 Västerås, Sweden; martina.summer.meranius@mdh.se; 4Christine E Lynn College of Nursing, Florida Atlantic University Boca Raton, FL 334 31, USA; cwill154@health.fau.edu; 5Department of Neurobiology, Care Sciences and Society, Karolinska Institutet, Alfred Nobels Allé 23, 141 52 Huddinge, Sweden

**Keywords:** spouse carers, older adults, caregiving, relationship, negative impact, positive value, support, dementia

## Abstract

(1) Background: Spouse carers of persons with dementia (PwD) are particularly vulnerable to negative outcomes of care, yet research rarely focuses on their caregiving situation. This study explores factors associated with the positive value and negative impact of caregiving in spouse carers of PwD in Sweden. (2) Methods: The study was a cross-sectional questionnaire-based survey, with a convenience sample of spouse carers of PwD (*n* = 163). The questionnaire addressed: care situation, carer stress, health and social well-being, relationship quality and quality of support, and contained measures of positive value and negative impact of caregiving. (3) Results: Hierarchical regression models explained 63.4% variance in positive value and 63.2% variance in negative impact of caregiving. Three variables were significant in the model of positive value: mutuality, change in emotional closeness following dementia and quality of support. Six variables were significant in the model of negative impact: years in relationship, years as carer, behavioural stress, self-rated health, emotional loneliness and change in physical intimacy following dementia. (4) Conclusions: Support to spouse carers of PwD should address the carer–care-recipient relationship quality, although different aspects of the relationship should be addressed if both the positive value of caregiving is to be enhanced and the negative impact reduced.

## 1. Introduction

Most care and support for persons with dementia (PwDs) is informal, i.e., provided by family or friends, usually an adult child or a spouse [1]. When compared to other informal carers, spouse carers are especially vulnerable to the negative effects of caregiving [2,3]. Numerous studies have focused on the negative effects of caregiving, while positive aspects have been relatively neglected, only being considered in a few primarily qualitative studies [4,5,6]. To effectively support spouse carers, it is important that the factors associated with both their negative and positive experiences of care are established to address the former and enhance the latter. This paper presents data from a cross-sectional survey of spouse carers of PwD in Sweden and explores factors associated with the positive value and negative impact of caregiving.

### 1.1. Factors Associated with Outcomes of Care

The level of impairment in PwD is associated with negative outcomes among carers, for example, higher levels of sleep disturbance, stress related conditions, and poorer health [3,7,8,9,10,11]. Other individual and contextual factors are also associated with poorer health outcomes in carers of PwD, including greater age of the carer, co-habitation with the care-recipient, and a longer duration of being a carer [10,12,13,14,15], all factors that characterise a spouse carer more so than a non-spouse carer. 

Spouse carers of PwD tend to take on more substantial care responsibilities than other carers [2], in some cases providing hours of care equivalent to or greater than full-time employment [2]. While being a spouse carer of PwD is associated with a variety of health outcomes, caring also affects their well-being and psychosocial situation. For instance, spouse carers of PwD experience higher levels of loneliness and social isolation than both non-spouse carers of PwD and other informal carers [16,17,18] and may withdraw from social situations after experiencing shame due to their partner’s dementia-related behaviours [19]. In addition, spouse carers of PwD can experience feelings that their future is uncertain, that their life lacks meaning, or be filled with grief due to the progressive loss of their partner to dementia [19,20].

### 1.2. The Importance of the Carer-Care Recipient Relationship for Outcomes of Care

Studies have highlighted the carer–care-recipient relationship as an important factor in the health and well-being of both parties. If the relationship is negatively affected as a consequence of a partner’s dementia, this can cause higher levels of grief and lower well-being in the carer [5,6]. However, studies have also found that a close relationship is associated with lower levels of burden, stress, and loneliness amongst spouse carers of PwDs [13,17,21].

The significance of the spousal relationship for positive outcomes of care is reflected in research that shows that being a spouse carer of PwD can be satisfying and provide a sense of meaning in everyday life [22,23,24]. Spouse carers of PwD who experience caring as burdensome can still be willing to continue caring and even increase the care they provide [2,25] and it has been suggested that this is due not only to insufficient formal support but also out of love and commitment [22,26].

The carer–care-recipient relationship has been explored using the concept of mutuality [26,27,28,29]. Mutuality can be defined as the positive aspects and qualities in a relationship, including dimensions of love, shared pleasurable activities, shared values, and reciprocity [30,31,32]. Reciprocity has been found to be a key part of, and contributor to, the experience of mutuality in dyadic relationships [31,33], where reciprocity is the equality in giving and receiving benefits or help between the two parties [27,34]. There is a lack of consensus regarding the primacy of the concepts of reciprocity and mutuality regarding caregiving relationships [28,31,33,34], and in the present study we examine mutuality and consider reciprocity a key component of mutuality.

For carers, the act of providing care is often embedded in and motivated by mutuality with longstanding reciprocal patterns in the relationship with the care-recipient. While some dimensions of mutuality in the current relationship might be limited due to the level of impairment in the care-recipient, the provision of care by the spouse can be seen as an act of love and piety but also as a way of reciprocating benefits received from the care-recipient in the past [26,27]. However, reciprocity does not need to be achieved through a like-for-like exchange, and the provision of support and care can be balanced by other qualities in the relationship, e.g., warmth and companionship. As such, a strong relationship with intimacy, emotional exchange and reciprocity can maintain and foster mutuality, and thus reduce carer burden and increase carer satisfaction [26,28,32].

### 1.3. The Present Study

Previous research has found that for support to carers to be optimally effective, it needs to be multimodal, focusing on several different factors related to the negative experience of care [35,36]. However, researchers have usually focused on the burdensome aspects of caregiving [4] and relatively few studies have investigated both negative and positive experiences of caregiving [37]. Thus, further research is needed to investigate those factors related to both the negative and positive aspects of caregiving, in order to develop an evidence-base for how to effectively reduce the former and promote the latter. Therefore, the aim of this study is to investigate which factors are associated with the negative impact and positive value of caregiving in spouse carers of PwD in Sweden.

## 2. Materials and Methods

### 2.1. Design

The design of the study was a cross-sectional questionnaire-based survey.

### 2.2. Sampling Frame and Participants

This study used convenience sampling via multiple recruitment channels (see Procedure) to recruit spouse carers of PwDs. Participants were individuals who identified with the definition given in the study information of a spouse carer of a PwD as someone “who provided care, help or support to a co-habitant spouse or partner with, or under evaluation for, a dementia disorder”. “Care, help or support” was further defined as “efforts a person makes on a regular basis such as personal care, supervision, household activities and maintenance, transportation, or contacts with services, this can include supporting the care-recipient’s personal economy, paying invoices, etc.”. In addition to being a spouse carer of a PwD, people were eligible for the study if they were: proficient in Swedish; aged 65 years or older; and living in ordinary housing.

A total of 175 individuals completed the study questionnaire of which 12 were excluded on eligibility criteria, providing an analytical sample of *n* = 163. As it is not possible to determine how many individuals were contacted in relation to the study, a response rate cannot be calculated.

### 2.3. Material

A questionnaire was developed by the research team. As far as possible, instruments and items validated in Swedish were used. Where there was no validated Swedish instrument, the instrument underwent standard translation procedures including back-translation from the original language. The questionnaire addressed the following areas: care situation; background characteristics; carer stress; health and social well-being; relationship quality; and quality of support. Items and instruments will be described consistent with how they were analysed, with variables used as outcome variables labelled dependent variables (DVs) and variables used as predictor variables labelled independent variables (IVs).

#### 2.3.1. Dependent Variables (DVs)

##### Care Situation

The COPE Index [38,39] is a 15-item measure of a carer’s perception of their caregiving situation, composed of three subscales, of which two were selected as DVs: Negative Impact (7 items, e.g., Does caregiving have a negative effect on your emotional well-being?) and Positive Value (4 items, e.g., Do you feel that anyone appreciates you as a caregiver?) of caregiving. All items have four response options ranging from “Never” (1) to “Always” (4). Cronbach’s alpha internal consistency reliability for the two subscales in our sample were: Negative Impact α = 0.82; Positive Value α = 0.70.

#### 2.3.2. Independent Variables (IVs)

##### Background Characteristics

Questions addressed the respondent’s and care-recipient’s gender and age, length of relationship and duration of care provision (in completed years).

##### Carer Stress

Level of stress due to providing care to a partner was measured by The Behavioural and Instrumental Stressors in Dementia (BISID) instrument [40], which measures the carer-reported (a) frequency of the care-recipient’s behavioural and instrumental problems and (b) level of stress experienced by the carer due to such problems. We report carer stress data only. In the behavioural section of BISID, the respondent is asked: “Does the person you care for behave in any of the following ways…” with a list of 12 behaviours. The instrumental section of BISID states: “The following questions are about the amount of personal care the person you care for needs. Could you please indicate how much help or supervision the person you care for needs with the following activities?”. Six activities are listed, with a further sub-section containing four questions on urinary and faecal incontinence by day and by night (combined into two (“by day or night”) for our study). Respondents rate the extent to which the behaviours and providing help with activities are stressful for them on a four-point response scale from “Not stressful” (0) to “Very stressful” (3). Sample Cronbach’s alpha for the two subscales: Behavioural Stress, α = 0.83; Instrumental Stress α = 0.89.

##### Health and Social Well-Being

Self-rated health was assessed with a single question for this purpose [41]: “How would you rate your general health?”, measured on a five-point response scale from “Excellent” (0) to “Poor” (4).

Sleep disturbance was measured by the question [2]: “During an average week, how much or little would you estimate that your sleep is disturbed because of the care and support you provide?”, with five response options ranging from “Not at all” (0), to “Every night” (4).

Loneliness was assessed by the 6-item de Jong Gierveld Loneliness Scale [42]. The scale comprises two subscales of social and emotional loneliness, each measured by three items describing examples of loneliness with response options “No”, “More or less” and “Yes”, coded 0, 1 and 1, respectively to indicate the absence or presence of loneliness. Sample Cronbach’s alpha for the two subscales: Emotional loneliness α = 0.57; Social loneliness α = 0.79.

Meaning in life was assessed using the Presence of Meaning subscale from the Meaning in Life questionnaire (MLQ) [43]. The subscale measures how full of meaning respondents feel their lives to be and contains five items phrased as statements with response options ranging from “Absolutely untrue” (1) to “Absolutely true” (7). Sample Cronbach’s alpha for the subscale: α = 0.87.

##### Relationship Quality

The Mutuality scale [44] contains 15 items capturing four dimensions of mutuality: love and affection; shared pleasurable activities; shared values; and reciprocity. Response options range from “Not close at all” (1), to “Very close” (4). Sample Cronbach’s alpha: α = 0.94.

Change in emotional closeness following dementia was measured by the question: ‘How emotionally close do you feel to your partner today compared with before she/he developed dementia?’, with response options “Less close” (−1), “Unchanged” (0), “More close” (1). Change in physical intimacy following dementia weas measured by the question: “How satisfied are you with your physical intimacy with your partner compared with before she/he developed dementia?”, with response options “Less satisfied” (−1), “Unchanged” (0), “More satisfied” (1).

##### Quality of Support

The previously described COPE Index [38,39] contains a third subscale, Quality of Support, consisting of four items measuring a respondent’s perception of how well supported they are as a carer, response options as for Negative Impact and Positive Value, above. Sample Cronbach’s alpha for the subscale: α = 0.65.

### 2.4. Procedure

In August 2019, a request to assist in the distribution of questionnaires was sent to a national network of social service carer support counsellors, dementia care providers, two dementia interest organisations and one family carer interest organisation. A total of 37 care providers and social service organisations and two civil society organisations responded to the request.

Care providers, social services and civil society organisations were instructed to identify and inform potential participants about the study and give them an envelope containing written information about the study, a consent form and the questionnaire. Parallel recruitment of study participants was conducted through social media platforms and news outlets. The questionnaire could optionally be completed via a paper version (89.0% of the analytic sample), a web-based version (9.8%) or by telephone interview (1.2%). Participants were instructed that the answers they gave in the questionnaire should exclusively concern the care they provided to their partner with dementia.

### 2.5. Data Analysis

SPSS v.28 was used for all analyses. For the BISID subscales and Mutuality scale, the means for the item scores were calculated to retain cases (BISID Behavioural and Instrumental Stress subscales, potential ranges 0–3, high score = high stress; Mutuality scale, potential range 0–4, high score = high mutuality). Summative scores were calculated for each of the three COPE Index sub-scales: Negative Impact, Positive Value and Quality of Support. Potential ranges for COPE Index sub-scales: Negative Impact 7–28, Positive Value 4–16 and Quality of Support 4–16, respectively. For the two subscales of the de Jong Gierveld 6-item Loneliness Scale, summative scores were calculated for Social Loneliness and Emotional Loneliness, both with potential range 0–3, high score = high loneliness. For the MLQ Presence of Meaning subscale, a summative score was calculated with potential range 5–35, high score = high presence of meaning.

Descriptive analyses as appropriate were performed on all study variables to examine central tendency and dispersion. Pearson’s product-moment correlation coefficient was estimated for bivariate associations between the study IVs and DVs.

Thereafter, two hierarchical regression models were developed, one for each DV. Order of entry of IVs in the models was determined to: first, control for background characteristics in the analyses (model step 1; only background characteristics with a significant bivariate association to at least one DV entered); second, examine whether carer stress, health and social well-being, and relationship quality variables would significantly improve the models at each step (steps 2, 3 and 4); and third, examine whether quality of support would significantly improve the models after inclusion of all other study variables (step 5). Pairwise deletion was employed for missing data to retain cases. Assumptions for multivariable analysis, including normality, linearity, homoscedasticity of residuals and the absence of collinearity and multicollinearity, were examined and found to be met. Level of significance for all analyses was set at *p* < *0*.05; due to multiple testing inflating the family-wise error rate each significance test should be considered in the context of the obtained effect size.

## 3. Results

### 3.1. Descriptive Analyses

The majority of spouse carers were female (76.7%), with a mean age of 75.3 years, while the majority of care-recipients were male (78.5%) with a mean age of 78.2 years (Table 1). Most spouse carers were married to their partner (91.0%) in a relationship that had lasted a mean of 48.6 years. Spouse carers had provided care for a mean of 4.40 years and their partner had received a dementia diagnosis a mean of 3.20 years previously.

Table 2 presents descriptive data on the remaining study variables. The mean score for Negative Impact was close to the scale midpoint, indicating that the average participant sometimes experienced negative impact. The mean score for Positive Value was slightly above the scale midpoint, indicating that the average participant often experienced positive value of caregiving. For most IVs, participants’ scores covered the potential ranges of the scales with mean scores close to the scale midpoints. The mean scores for the Behavioural and Instrumental Stress subscales were both below their scale midpoints, with the mean for Behavioural Stress slightly higher. This indicates that the average participant experienced no to little stress due to the care-recipient’s behavioural and instrumental problems. The average participant was less emotionally close to their partner and experienced reduced satisfaction with physical intimacy than prior to their partner developing dementia.

### 3.2. Bivariate Analyses

The two DVs in the study, Negative Impact and Positive Value, had a significant negative correlation (*r* (157) = −0.415, *p* ≤ 0.001). Table 3 presents the correlation coefficients between the study IVs and each DV. All IVs were significantly associated with Negative Impact, with the strengths of the correlations ranging from *r* = −0.16 for carer age to *r =* 0.62 for Behavioural Stress.

Several variables were not significantly associated with Positive Value: carer gender, years in relationship, years as a carer, and Instrumental Stress and sleep disturbance. For the IVs that were significantly associated with Positive Value, the strengths of the correlations ranged from *r* = −0.18 for self-rated health to *r* = 0.70 for Mutuality.

### 3.3. Multivariable Analyses

In accordance with the principle of parsimony, carer age was omitted from the multivariable models due to its significant association with years in relationship (*r* (156) = 0.483, *p* ≤ 0.001) and a preference for examining how years in relationship might be related to Negative Impact and Positive Value in the models.

Table 4 displays the unstandardized coefficients (*B*), standard error of *B*, standardized coefficients (*β*) and significance (*p*) of the IVs for the final hierarchical regression models for Positive Value and Negative Impact.

In the multivariable model of Negative Impact, *R* was significantly different from zero at each step and for the final model: *R* = 0.795, *F* (13,133) = 17.57, *p* ≤ 0.001. As Quality of Support (step 5) did not produce a significant increment in *R*^2^ compared to the model at the end of step 4, we report the data for step 4 below and in Table 4. Six IVs were significant in the model at the end of step 4: years in relationship (*sr*^2^ = 0.03), years as carer (*sr*^2^ = 0.04), Behavioural Stress (*sr*^2^ = 0.05), self-rated health (*sr*^2^ = 0.03), Emotional Loneliness (*sr*^2^ = 0.04) and change in physical intimacy (*sr*^2^ = 0.02).

In the multivariable model of Positive Value, *R* was significantly different from zero after each step. With all IVs entered in the model, *R =* 0.796, *F* (14,132) *=* 16.30, *p* ≤ 0.001. Three IVs were significant in the final model. The unique variance (*sr*^2^) in the DV explained by these significant IVs were: Mutuality (*sr*^2^ = 0.10), Change in emotional closeness (*sr*^2^ = 0.01) and Quality of Support (*sr*^2^ = 0.06).

## 4. Discussion

The present study aimed to investigate factors associated with negative impact and positive value of caregiving for spouse carers of PwD. At bivariate level, the study IVs had inverse associations with the measures of negative impact and positive value, while analyses at the multivariable level showed that no IV was significant in both positive value and negative impact models. Combined with the finding that the measures of negative impact and positive value were only moderately correlated with a shared variance of 17%, there is little evidence to suggest that the positive value a carer gains from providing care is simply the opposite or absence of the experience of negative impact. Rather, this study supports previous research indicating [36] that negative impact and positive value are two related but divergent experiences of caregiving and should thus gain equal attention in research and practice.

The multivariable model of negative impact presented a rather complex pattern of association between IVs and DV. With the significant variables in the model explaining from 2–5% unique variance, no single variable stood out as a primary predictor. Stress due to behavioural problems in the care recipient, years as a carer and self-rated health were significant predictors of negative impact, which taken together could suggest that the negative impact of caregiving is influenced by increasing stress as the care-recipient’s behavioural problems worsen combined with deteriorating health in the carer. However, research has found that negative outcomes of being a spouse carer do not worsen over time but tend to decrease after 4 to 7 years [10]. The carers in our sample had been providing care for a mean of 4.40 years, with their partner’s dementia diagnosed a mean of only 3.20 years previously. It is therefore possible that many carers in our sample had not had time to develop coping strategies to manage the care-recipient’s dementia.

In contrast to how the duration of providing care was associated with negative impact, the length of the spousal relationship had a protective effect. Research has found that a long marital relationship allows bonds to develop that provide the spouse carer and care-recipient with the resilience to face their “new reality” with dementia together [45]. Studies have shown that poorer relationship quality and reduced levels of intimacy in caregiving relationships are linked to loneliness [23,24], and in our analyses both emotional loneliness and a reduction in physical intimacy in the spousal relationship following dementia were associated with negative impact.

In the multivariable model of positive value, mutuality stood out as the IV explaining most variance in the DV and that experiencing more positive value in caregiving was associated with both higher levels of mutuality and an increase in emotional closeness to their partner following dementia. Other studies have shown that a close relationship and mutuality between partners can foster both a more positive caregiving situation amongst spouse carers and reduce negative outcomes [5,21,31]. This may however be an oversimplification, as the present study found that those aspects of the carer-care recipient relationship associated with positive value were not the same aspects as those associated with negative impact. For example, while mutuality explained 10% unique variance in positive value, it was not a significant predictor in the model of negative value, while a loss of physical intimacy following dementia was. While both multivariable models point to the importance of relationship quality for the experience of caregiving, the contrasts between the models would suggest that different aspects of relationship quality are related to negative impact from those related to positive value.

Several studies have concluded that a “one-size-fits-all” approach to carer support will not meet the needs of carers in general nor those of specific carer groups such as spouse carers [3,20,46,47,48]. Still the support offered is for the most part generic, primarily carer training, counselling, or support groups, and not predicated on the carers’ individual needs or preferences [49,50]. A surprising finding of the present study was that quality of support was not significant in the model of negative impact. This could indicate that the support currently provided to spouse carers of PwD—whether from formal services or from family and friends—is not adequate to meet their complex needs and to reduce negative impact by, e.g., alleviating the stress from the care-recipient’s behavioural problems, counteracting emotional loneliness, and improving physical intimacy. However, while quality of support failed to significantly improve the model of negative impact, the variable significantly predicted 6% unique variance in positive value. One interpretation of the findings of the present study is that being supported, even at relatively low levels or in a generic fashion, may help a carer gain some satisfaction from caring [51], but that in order to reduce negative impact the support needs to be more substantial and/or less generic.

The findings from the present study would indicate that support for spouse carers of PwD needs to maintain and strengthen their relationship to the care-recipient. Arguments have been made that support should target factors associated with carer burden, while also aiming to strengthen the carer’s resources [10,24,51,52,53]. Among spouse carers, studies suggest that support that aims to promote continuity in and strengthen the relationship with the care-recipient might increase well-being in both parties [6,21,54]. An example of such support is the provision of meaningful activities in which both spouse carer and their partner can engage [21,55,56]. Research indicates that spouse carers of PwD often experience shame due to their partner’s dementia when in public [16,19]. By providing opportunities for carer and care-recipient to, e.g., meet with other carers and PwD for meaningful activities in safe environments, carers can continue support their partner while the emotional loneliness associated with their role can be reduced and the mutuality and reciprocal patterns developed over time with their partner can be maintained.

In addition, spouse carers of PwD might need individual support targeting complex and sensitive issues such as physical intimacy and emotional closeness with their partner, while other actions might address the PwDs behavioural problems. Thus, support needs to be individualized and adaptive from earlier to later stages of caring. Support also needs to be flexible and responsive as unknown issues may develop over the length of the caregiving relationship related to, e.g., changes in the spouse carer’s own health and living situation.

## 5. Study Strengths and Limitations

An original feature of this study is its focus on both negative and positive outcomes of care. Among the strengths of the study are its specific focus on spouse carers of PwD combined with a broad perspective on their caregiving situation encompassing their perceptions of the care-recipient’s behavioural problems and instrumental needs, their own health and social well-being, and the quality of their relationship with the care-recipient. This provided the foundation for new insights on the complex situation for spouse carers of PwD.

Identifying and recruiting a sample of spouse carers of PwD is challenging and required support of several actors and organisations within a convenience sampling strategy. As such, the relatively small convenience sample of carers in the study means that generalisation of its findings to the wider population of spouse carers of PwD should be done with caution. The sample might be biased toward those spouse carers of PwD who are in contact with service providers, with an underrepresentation of those more isolated in their carer role.

Sample size, and the number of scale items will all influence Cronbach’s alpha for internal consistency reliability. Some scales used in this study had relatively low Cronbach’s alpha, although only one—the emotional loneliness subscale—had an alpha level commonly regarded as poor. For scales with questionable or poor alpha, average inter-item correlations were analysed and found to be satisfactory.

## 6. Conclusions

This study highlights the centrality of the relationship between spouse carers and PwD when considering outcomes of caregiving. However, caregiving relationships are complex, and our findings indicate that those aspects of the relationship associated with the positive value of care differ from those associated with its negative impact. We conclude that support to spouse carers of PwD needs to sustain the relationship between carer and care-recipient in a variety of ways in order to reduce the negative experience of care and enhance its positive value.

## Figures and Tables

**Table 1 ijerph-19-01788-t001:** Sample characteristics and their care-recipients (*N* = 163).

Variable		
Gender. Spouse carer		
	Female (%)	76.7
Gender. Care-recipient.		
	Female (%)	21.5
Age. Spouse carer (*n* = 161)		
	Mean (SD), range	75.30 (5.82), 65–89
Age. Care-recipient (*n* = 160)		
	Mean (SD), range	78.22 (6.67), 62–93
Relationship to care-recipient. (*n* = 155)		
	Married (%)	91.0
	Partner (%)	9.0
Years in relationship (*n* = 160)		
	Mean (SD), range	48.61 (13.44), 8–70
Years of co-habitation (*n* = 156)		
	Mean (SD), range	46.30 (13.23), 7–69
Years as carer (*n* = 156)		
	Mean (SD), range	4.40 (4.52), 1–43
Years since received dementia diagnosis (*n* = 152)		
	Mean, (SD), range	3.20 (2.92), 1–20
Dementia diagnosis received (*n* = 159)		
	Alzheimer’s disease (%)	49.1
	Vascular dementia (%)	19.5
	Dementia with Lewy bodies (%)	5.7
	Frontotemporal dementia (%)	3.1
	Dementia due to Parkinson’s disease (%)	1.9
	Unspecified or mixed dementia (%)	20.8

**Table 2 ijerph-19-01788-t002:** Descriptive statistics for care situation, health and well-being, and quality of relationship variables.

Variable	*M*	*SD*	Range	*n*
Negative impact ^a^	15.31	3.90	7–28	159
Positive value ^a^	11.47	2.48	4–16	161
Behavioural stress ^b^	0.89	0.52	0–2.45	156
Instrumental stress ^b^	0.64	0.67	0–2.75	158
Self-rated health	2.39	0.95	0–4	160
Sleep disturbance	1.63	1.38	0–4	160
Social loneliness ^c^	1.68	1.24	0–3	158
Emotional loneliness ^c^	1.57	0.97	0–3	157
Meaning in life ^d^	23.89	6.58	5–35	155
Mutuality ^e^	2.52	0.66	1.07–3.73	158
Change in emotional closeness	−0.48	0.69	−1–1	159
Change in physical intimacy	−0.75	0.46	−1–1	159
Quality of support ^a^	9.86	2.76	4–16	161

Note: ^a^, COPE Index; ^b^, BISID; ^c^, de Jong Giervald loneliness scale; ^d^, Meaning in life, Presence of meaning subscale; ^e^, Mutuality scale.

**Table 3 ijerph-19-01788-t003:** Bivariate correlations between independent and dependent variables.

Variable	Negative Impact ^a^		Positive Value ^a^	
	*r*	*p*	*r*	*p*
Age. Spouse carer	−0.161	0.044	0.207	0.009
Gender. Spouse carer	0.174	0.029	−0.154	0.051
Years in relationship	−0.184	0.022	0.152	0.056
Years as carer	0.195	0.016	−0.116	0.155
Behavioural stress ^b^	0.616	<0.001	−0.405	<0.001
Instrumental stress ^b^	0.337	<0.001	−0.069	0.389
Self-rated health	0.376	<0.001	−0.177	0.026
Sleep disturbance	0.405	<0.001	−0.078	0.329
Social loneliness ^c^	0.252	0.001	−0.270	0.001
Emotional loneliness ^c^	0.435	<0.001	−0.332	<0.001
Meaning in life ^d^	−0.334	<0.001	0.464	<0.001
Mutuality ^e^	−0.512	<0.001	0.700	<0.001
Change in emotional closeness	−0.330	<0.001	0.438	<0.001
Change in physical intimacy	−0.398	<0.001	0.270	0.001
Quality of support ^a^	−0.291	<0.001	0.530	<0.001

Note: ^a^, COPE Index; ^b^, BISID; ^c^, de Jong Giervald Loneliness Scale; ^d^, Meaning in life, Presence of meaning subscale; ^e^, Mutuality scale; For analyses *n* varies between 151 and 161 due to internal missing values.

**Table 4 ijerph-19-01788-t004:** Hierarchical regression models for Positive value of caregiving and Negative impact of caregiving (*n* = 146).

Variable	Negative Impact				Positive Value			
	*B*	*SE B*	*β*	*p*	*B*	*SE B*	*β*	*p*
Background characteristics								
Gender	0.615	0.501	0.067	0.222	−0.118	0.320	−0.020	0.714
Years in relationship	−0.053	0.017	−0.184	0.002	0.002	0.011	0.010	0.860
Years as carer	0.183	0.050	0.212	<0.001	−0.034	0.032	−0.061	0.294
Carer stress								
Behavioural stress	2.220	0.544	0.298	<0.001	−0.364	0.348	−0.077	0.298
Instrumental stress	0.168	0.364	0.029	0.645	0.318	0.233	0.085	0.175
Health and social well-being								
Self-rated health	0.871	0.259	0.211	0.001	0.055	0.166	0.021	0.738
Disturbed sleep	0.213	0.184	0.075	0.250	0.032	0.119	0.018	0.790
Social loneliness	0.090	0.195	0.029	0.644	−0.031	0.142	−0.015	0.830
Emotional loneliness	1.041	0.259	0.259	<0.001	0.002	0.166	0.001	0.992
Meaning in life	−0.031	0.038	−0.053	0.418	0.038	0.025	0.101	0.122
Quality of relationship								
Mutuality	−0.157	0.455	−0.026	0.731	1,751	0.292	0.460	<0.001
Change in emotional closeness	−0.092	0.372	−0.016	0.804	0.544	0.238	0.152	0.024
Change in physical intimacy	−1.454	0.514	−0.173	0.005	−0.069	0.330	−0.013	0.836
Support								
Quality of support ^a^	-	-	-	-	0.280	0.061	0.312	<0.001
Regressions	*R* ^2^	*R* ^2^ * _adj_ *	*F_inc_*	*p*	*R* ^2^	*R* ^2^ * _adj_ *	*F_inc_*	*p*
Step 1 ^i^	0.112	0.094	(3,143) = 6.04	0.001	0.064	0.044	(3,143) = 3.25	0.024
Step 2 ^ii^	0.432	0.412	(2,141) = 39.67	<0.001	0.205	0.176	(2,141) = 12.48	<0.001
Step 3 ^iii^	0.602	0.573	(5,136) = 11.63	<0.001	0.387	0.342	(5,136) = 8.12	<0.001
Step 4 ^iv^	0.632	0.596	(3,133) = 5.59	0.016	0.575	0.534	(3,133) = 19.59	<0.001
Step 5 ^v^	-	-	-	-	0.634	0.595	(1,132) = 21.06	<0.001

Note: Table presents data for final models; ^a^ Final model data presented for Negative Impact after step 4 as Quality of support did not produce significant increment in model at step 5; ^i^ Background characteristics; ^ii^ Background characteristics, Carer stress; ^iii^ Background characteristics, Carer stress, Health and social well-being; ^iv^ Background characteristics, Carer stress, Health and social well-being, Quality of relationship; ^v^ Background characteristics, Carer stress, Health and social well-being Quality of relationship, Quality of support; Due to missing values *n* = 146 for both models.

## Data Availability

The data presented in this study are available on request from the corresponding author. The data are not publicly available due to restrictions in accordance with the Swedish Public Access to Information and Secrecy Act (2009:400). Requests for access to the dataset should be sent to the corresponding author, and will be considered by the University’s data protection officer.

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
