# Peer review of "Negative Impact and Positive Value of Caregiving in Spouse Carers of Persons with Dementia in Sweden"

_ijerph, 2022, doi:10.3390/ijerph19031788_

Round 1
Reviewer 1 Report
See the attachement, please

Author Response
Response to reviewer 1
Thank you for submitting your paper for review and for conducting the study. This study examined factors associated with the positive value and negative impact of caregiving in spouse caregivers of PwD in Sweden. There is interesting data in this study, the authors have conducted the study with focus on high methodological standards to provide a valuable opportunity in measuring care situation, carer stress, health and social well-being, relationship quality and quality of support, and contained measures of positive value and negative impact of caregiving.
Response:
Thank you for your thorough review of our manuscript and for your appreciative comments.
Comment:
In the introduction, provide sufficient background and include more relevant references and gap in the literature to provide the rationale for this study.
Response:
As the reviewer does not indicate in what way the background to the study provided in the Introduction is insufficient, it is not clear how we should respond to this comment. However, we have now extended and re-written the Introduction sub-section ‘The present study’ to more precisely describe the current research gap and to connect this to the rationale for a study that examines both the negative and positive experiences of caregiving, and those factors associated with both. We have also added in three new references to studies that address how to support carers of people with dementia and/or positive aspects of caregiving (lines 104-112).
Comment:
The methods section is detailed and explain all the relevant information for this type of study that can be replicated. Design is appropriate.
Response:
Thank you. Following comments from Reviewer 2, we have made minor clarifications to the data analysis section.
Comment:
Result Section. I find it easy to understand how interpret the data and what are the Tables are showing.
Response:
Thank you. This section has not been revised.
Comment:
Discussion Section. It starts the discussion reporting the own findings from the present study and then, after that, you put it in perspective of other available research. Please, only include a clinical implications section more.
Response:
We agree that it is important to discuss the clinical implications of our study. However, we have followed the journal author guidelines which state that implications should be discussed ‘in the broadest context as possible’ and not as a separate heading or within a narrow focus. Therefore, in the present discussion, we discuss a range of implications in relation to our findings on lines 411-423.
Comment:
Conclusion Section. The conclusion should state only your findings.
Response:
Overall, we believe that the conclusions as written derive from our findings. However, following this comment we have revised the conclusions by removing the last sentence, which was calling for further research on a related topic (lines 454-457).
Reviewer 2 Report
The article deals with a widely studied topic, the care relationship with a spouse with dementia, but uses an original approach that considers the negative and positive aspects of this caring relationship separately.
As a premise, I must congratulate you on the interesting study and the results, as well as on the clarity of the method of analysis adopted. The introduction correctly reports the literature data on the topic of the article, useful for better understanding the question that the research would like to answer. In the method section, the procedures adopted and the statistical methods are explained logically and clearly. Results well support the conclusions; their meanings are properly analyzed in the discussion section.
I find that an interesting further topic of analysis and discussion could have been the influence of gender, which has not been considered. Is there a reason for this?
Some detailed comments:
54 Spouse carers of also take on more …. I believe the sentence needs to be modified
172 – 3 : Loneliness was assessed by the 6-item De Jong Gierveld Loneliness Scale [39]. The scale comprises two subscales of social and emotional loneliness, each measured by three items describing examples of loneliness with response options ‘No’, ‘More or less’ and ‘Yes’, coded 0, 1 and 1 respectively to indicate the absence or presence of loneliness….: it should be pointed out that the relation between the three options with the point 0 – 1 may change as the meaning of the question change.
222 For the remaining instruments, scale/sub-scale items were summed, potential ranges:.. I suggest rewriting the sentence more clearly
261 Negative Impact was below the scale midpoint : in Tab 2 the mean value was 15.31 while the range was 7 – 28; it does not seem below the midpoint.
Table 4. Hierarchical regression models : Could you find a way to remind readers that each step refers to each subsequent area in the table?
Author Response
Response to reviewer 2
Comment:
The article deals with a widely studied topic, the care relationship with a spouse with dementia, but uses an original approach that considers the negative and positive aspects of this caring relationship separately.
As a premise, I must congratulate you on the interesting study and the results, as well as on the clarity of the method of analysis adopted. The introduction correctly reports the literature data on the topic of the article, useful for better understanding the question that the research would like to answer. In the method section, the procedures adopted and the statistical methods are explained logically and clearly. Results well support the conclusions; their meanings are properly analyzed in the discussion section.
I find that an interesting further topic of analysis and discussion could have been the influence of gender, which has not been considered. Is there a reason for this?
Response:
First of all, thank you for reviewing our manuscript and your appreciative comments.
We agree that gender is an important issue when considering informal care. We chose not to make gender a main focus in this paper as most (over 75%) of the spouse carers in our sample were female. Stratifying our analyses by gender would have created problems for the multivariable model of male spouse carers due to the cases-to-variables ratio. We did consider gender as a background variable in our analyses but as it had weak bivariate and non-significant multivariable associations with the two dependent variables, we chose not to further elaborate on this variable in the discussion but to focus on our significant results. However, we agree that it would be interesting to further investigate how gender could impact positive value and negative impact of care and will consider this for future studies where a larger sample would allow for gender-stratified analyses.
Detailed comments
Comment:
54 Spouse carers of also take on more …. I believe the sentence needs to be modified
Response:
54: We agree, the sentence could be interpreted as a subordinate clause which was the case in an earlier version of the manuscript. We have revised the sentence to better accommodate its position in the paragraph (lines 56-59)
Comment:
172-3: Loneliness was assessed by the 6-item De Jong Gierveld Loneliness Scale [39]. The scale comprises two subscales of social and emotional loneliness, each measured by three items describing examples of loneliness with response options ‘No’, ‘More or less’ and ‘Yes’, coded 0, 1 and 1 respectively to indicate the absence or presence of loneliness….: it should be pointed out that the relation between the three options with the point 0 – 1 may change as the meaning of the question change.
Response:
You raise a valid point, and additionally there does seem some redundancy in first requiring a response on a three-point scale when this information is then lost when the scale is converted to a dichotomous (presence/absence) measurement. Nevertheless, the coding scheme you refer to is that recommended by the instrument developers and the instrument has been validated in several studies and in a number of languages. We do note in the Strengths and limitations section of the Discussion that the Cronbach’s alpha for the emotional loneliness subscale was poor (lines 440-443), and this may reflect the issue you raise; however, as also stated, we did carry out additional checks on the subscale (and the other scales in the study). While sharing your reservations about the instrument, we do not think there is much to be gained in expanding on our comments in the current version of the paper.
Comment:
222 For the remaining instruments, scale/sub-scale items were summed, potential ranges:.. I suggest rewriting the sentence more clearly
Response:
222: We agree that this sentence lacked some clarity and have now been rewritten in order to make the information clearer (lines 234-241).
Comment:
261: Negative Impact was below the scale midpoint: in Tab 2 the mean value was 15.31 while the range was 7 – 28; it does not seem below the midpoint.
Response:
Thank you for pointing this out this error. It should read that Negative impact was close to scale midpoint. This has now been corrected (line 280).
Comment:
Table 4. Hierarchical regression models: Could you find a way to remind readers that each step refers to each subsequent area in the table?
Response:
A note detailing the variables included in each step of the analysis has been added to the table (lines 323-327).